# Individuals with and without Visual Impairments Use a Force Feedback Device to Identify the Friction and Hardness of Haptic Surfaces

**DOI:** 10.3390/s22249745

**Published:** 2022-12-12

**Authors:** Konstantinos Papadopoulos, Eleni Koustriava, Evangelia Georgoula, Vaia Kalpia

**Affiliations:** Department of Educational and Social Policy, University of Macedonia, 54636 Thessaloniki, Greece

**Keywords:** haptic devices, surface friction, surface hardness, visual impairments

## Abstract

The general purpose of this study is to promote access to haptic virtual environments. Using a haptic device, people with and without visual impairments (VI) are able to feel different textures and compare these textures based on different surface properties, i.e., friction and hardness. The objectives of this study were to examine the following: (a) whether the variables of friction and hardness were identifiable through the Touch device (Phantom Omni) and could therefore function as 3D haptic variables; (b) if there were differences between people with VI and sighted individuals in terms of their performance; (c) the differences that should exist between the values of each variable so that the virtual surfaces could be identified as different to each other; and (d) if the individual characteristics of participants have an impact on their performance. The results showed that it is necessary to use surfaces which are differentiated based on the degree of friction and hardness because the haptic properties of a virtual object are then better perceived. Individuals with VI need more time and more effort to understand friction and hardness, respectively. With the motivation of increasing access to object perception for people with VI in a virtual environment, accessibility advisors and experts can extract useful information for the development of functional and efficient 3D objects for haptic perception.

## 1. Introduction

Even nowadays, the use of graphic communication has excluded individuals with VI [1], as they are deprived of visual information [2]. However, innovative technology and software can transfer information not only by auditory and visual means, but also by adding the sense of touch [3,4].

Touch is important in our everyday interactions, and allows perception of, and interaction with, objects [5]. Object information can be separated into both geometrical (such as shapes) and material properties [6,7,8,9], with variability in them directly affecting identification [10]. Object recognition studies have put emphasis on the shape rather than material and surface properties [11,12]. The surface properties of an object can both provide useful information on the object’s identity [11] and increase the naturalness of virtual environments [13].

A basic distinction between tactile (the physical environment) and haptic (the virtual environment) perception should be made. During tactile perception, people have the ability to use their fingers directly on a stimulus and explore it in detail. However, when objects are placed between the fingers and the stimulus, fewer features can be perceived [7]. Thus, in the case of haptic perception which involves the use of a force feedback device, a stylus is placed between the fingers and the stimulus, resulting in the user touching only one point of the 3D object at a time [14,15]. In this case, it has been suggested that only specific features of the stimulus, such as vibration and roughness, can be perceived by moving the handle towards or across the virtual surface [16].

Three subjective dimensions and specific features—roughness, hardness, and stickiness/slipperiness—dominate the haptic perception of surfaces [17]. Roughness is another dimension of surfaces and is formed mainly by differentiating the height or/and the pattern of the elements on a surface, as well as the distance among them [7,18]. In addition, changes in the friction of a virtual surface can change how rough this surface feels [19]. Friction is defined by stickiness and slipperiness and is one dimension of a surface [20]. Finally, hardness is defined by the degree of intrusion of one object (for instance, the finger or the virtual cursor) into another (a physical object or a virtual object with respect to the previous example).

Only one of these dimensions—roughness—has been widely studied [20,21]. The dimension of roughness has been studied by many researchers using haptic feedback devices [14,18,19,22,23,24,25,26,27,28]. Most of these studies included sighted individuals, apart from the ones by Jansson et al. [14] and Kornbrot et al. [26] in which blind and sighted individuals participated. The findings of these studies support the perception of roughness in virtual surfaces with and without vision. More specifically, Kornbrot et al. [26] found that status of vision was not correlated with roughness perception and Jansson [23] found that the blindfolded participants in his research performed well, suggesting that blind individuals would have performed equally well or perhaps even better. However, when he conducted a similar study including both individuals with blindness and sighted individuals, he concluded that there were differences in performance between the two groups [14,24].

The second major dimension—hardness—has been much less studied than that of roughness in a virtual environment. Few studies, such as [29,30,31,32], have dealt with the perception of hardness simulated by haptic feedback devices. Specifically, the devices used in these studies were the Phantom Omni, a one degree of freedom force display, the Phantom, and the Freedom6S hand controller, respectively. Additionally, people with VI did not participate in these studies.

Regarding the third dimension of surface haptic perception, stickiness and slipperiness are related to the dimension of friction [20]. Few studies, such as [21,33,34], have investigated the perception of friction using haptic feedback devices, especially the Phantom device used by participants without VI in these studies.

Evidently, people with VI have not participated in the majority of the aforementioned studies. As technology is advancing at a rapid rate, it is increasingly important to consider the different capabilities of many users [35]. Haptics in virtual environments enables users, including individuals with VI, to feel, touch, and interact with virtual objects in such environments [35]. Given that haptics in these environments can be implemented in various fields, such as education, cartography, games, and cultural heritage, it is important to ensure accessibility to them. For example, virtual reality applications are commonly used in museums, but the major disadvantage of them is that they do not allow visitors to experience the tactile exploration of cultural heritage artifacts [36], and so they are still far from accessible to users with VI.

The general idea of this study is to explore how to improve accessibility to a haptic virtual environment. For example, how can people with VI perceive the artifacts/objects of a museum if tactile exploration is not allowed? In this case, the perception of artifacts can become available through haptic devices, such as the Haption Virtuose 6D Force Feedback Device, the Geomagic Touch X (formerly Sensable Phantom Desktop), the (formerly known as) Phantom Omni Force Feedback Haptic Device (now “Touch”), or NOVINT Falcon 3D Force Feedback Haptic Device.

However, the use of haptic devices with digital 3D object models cannot provide the information needed to fully and accurately identify and understand 3D objects. The application of 3D haptic variables/properties to digital 3D models through properly designed applications could fill this gap. Thus, the basic concept of this study is the identification of 3D haptic variables which could be applied to the construction of 3D objects that a user would sense through a haptic feedback device. For instance, the recognition of a 3D artifact that represents a human face could be enabled and finally achieved by applying different textures on the surfaces. The differentiation of textures could be based on the use of different haptic variables (e.g., roughness or hardness) or the use of the same haptic variable but with different values (e.g., hardness of higher and lower value) so that the facial features could be sensed differently. Although 2D haptic variables have been referred to previously, such as in [37] (p. 94), 3D haptic variables/properties have not yet been specified. The present study is the first in a series of surveys that we plan to conduct in the coming years to determine 3D haptic variables for different haptic devices.

The general objective of the present study was to investigate the appropriateness of utilizing the 3D haptic properties/variables of friction and hardness to develop differently sensed surfaces identifiable through a haptic feedback device. In the present study, friction and hardness were examined using the same and different surfaces with regard to their exponent, i.e., the value that defines the degree of friction or hardness. The specific objectives of the study were to examine the following: (a) whether the variables of friction and hardness were identifiable through the Touch haptic device (Phantom Omni), and could therefore function as easily identifiable 3D haptic variables; (b) if there were differences between people with VI and sighted individuals in the identification of friction and hardness through the Touch haptic device; (c) the differences that should exist between the values of each variable (friction and hardness) so that the virtual surfaces can be identified as different to each other (e.g., the user of the haptic device should feel that they were touching surfaces with different levels of friction); and (d) if the individual characteristics of participants have an impact on their performance in the experiment.

## 2. Materials and Methods

### 2.1. Participants

A total of 88 young and middle-aged adults participated in this experiment. In total, 45 of the participants were adults with VI (28 participants with blindness, 6 with severe VI, and 11 with low vision) aged between 19 and 56 years old (M = 32.6, SD = 11.2) and 43 were sighted (blindfolded) and aged between 19 and 57 years old (M = 32.2, SD = 10.3). The mean age at onset of VI was 4.89 (SD = 6.94), and the mean age of vision loss was 11.1 (SD = 9.11). In addition, the years living with VI were calculated (age minus age at onset of VI) (M = 27.7, SD = 13.5). None of the participants had cutaneous or kinesthetic impairments.

In terms of the level of education of individuals with VI, 16 participants were university students, 16 were university graduates, 9 had graduated from high school, 2 had graduated from junior high school, 1 was an adult high school student, and 1 had graduated from an institute of vocational training. With regard to the educational level of the sighted participants, 20 were university graduates, 9 were university students, 9 had a master’s degree, 3 had graduated from high school, and 2 had a Ph.D.

Fourteen individuals with VI had previous experience with similar haptic feedback devices, but none of the sighted participants had any similar experience before the experiment. Moreover, participants with VI were asked to refer to the reading media they used. Thirty-four participants stated that their reading media was Braille or audio-supported reading, three said that they used (magnifier) books with low vision aids, and eight stated that they used both Braille/audio-supported reading and (magnifier) books with low vision aids.

### 2.2. Instruments and Procedures

The haptic feedback device used for the experiment has three degrees-of-freedom in terms of force feedback and six-degrees-of-freedom in terms of positional sensing, making it appropriate for diverse environments [38]. The device integrates a pen-like handle as the end effector. Thanks to simulation software, it is possible to create an interaction for the user involving virtual surfaces with different properties. This device was formerly known as the Phantom Omni, but it has since been named (Geomagic) Touch [39]. It is the most widely used haptic feedback device [40,41] and one of the best on the market [40]. Additionally, the Touch has higher resolution than other haptic devices, such as the Novint Falcon [38]. The relative ease of use and the high resolution of the device in virtual environments made the Touch the ideal device for the purposes of this study.

The tests were presented on a laptop using the H3DAPI program, which depicts three-dimensional models. In order to enhance the digital surfaces with tactile properties, the authors used Haptics3D (H3D) [42], which is a piece of open-source software written in C++. H3D supports Python and C++ programming. It uses the Open Graphics Library (OpenGL) API for rendering 2D and 3D graphics. ISO-ratified eXtended 3D (X3D) was used to represent 3D computer graphics. The H3D files were written in XML syntax with an X3D extension. Each 2D and 3D object in the H3D is referred to as a node. Nodes are the basic element in H3D. Haptic and tactile attributes such as stiffness and static and dynamic friction can be assigned to each node with the help of xml tags.

The experiment took place in a quiet room. Each participant sat comfortably in front of the Touch (see Figure 1). Sighted and low-vision participants were blindfolded during the experiment. Before starting the experiment, participants were informed of the safety of the device and told how to use the stylus connected to the haptic feedback device. They were also informed that exploring a virtual surface through a haptic device was similar to using a rigid tool (such as a pen) as a mediator. A phase of familiarization with the device was followed by the administration of two pilot tests created for this practice session. The time limit was five minutes for each test.

After the familiarization phase, the two experimental tests were carried out. In test 1, the haptic variable of friction was examined. In test 2, it was the haptic variable of hardness which was investigated. Each test was divided into four surfaces: upper-left, upper-right, bottom-left, and bottom-right (see Figure 2). Two out of the four surfaces had the same values, which meant that the user had the same feeling when using the haptic feedback device.

The values given for the test of friction were 0.00, 0.35, and 0.70. Static and dynamic friction had the same values. The value 0.35 was given twice, which means that two out of the four virtual surfaces were the same. The values for the test of hardness were 0.35, 0.70, and 1.00. The value 0.70 was given twice which, again, means that two out of the four virtual surfaces were the same.

The participants carried out six trials for each test. The six trials were as follows: (1) upper-left and upper-right virtual surface, (2) bottom-left and bottom-right, (3) upper-right and bottom-right, (4) upper-left and bottom-left, (5) upper-left and bottom-right, and (6) upper-right and bottom-left. When the six trials were completed the test was finished.

On every trial of the experiment, the participants “were touching” the two surfaces through the haptic device and stated whether the two surfaces felt the same or different. The researcher noted correct and incorrect answers. The answers of the participants were “yes” if the two surfaces were the same and “no” if the two surfaces had differences. For each trial, there was only one try and a time limit. The minimum and maximum time limit allowed was from 30 to 60 s. This means that the participants had to explore the surfaces for at least 30 s. Next, they had two choices—either respond or continue exploring the surfaces until they were ready to respond, but no later than 60 s after the beginning of the trial. When the 60 s had passed, participants were obliged to give an answer as no additional time could be granted.

The tests were presented to the participants in a cyclical process. For example, the first participant started from test 1, the second participant from test 2, the third participant from test 1, and so on. This process was chosen in order to exclude the possibility of a participant’s fatigue influencing the results obtained.

## 3. Results

Initially, the means of the participants’ scores (i.e., the ratio of correct responses with respect to the number of trials) in each test were calculated, and then the means of the participants’ completion time (i.e., the time they needed to complete a trial in each test) in seconds were calculated as well (see Figure 3).

Six new variables were formed to examine the possible effect of (1) the identification of the same versus different (in texture) surfaces that had been developed with the use of the same or different values for the variables “friction” and “hardness” and (2) the identification of surfaces that presented values different to each other for the variables friction and hardness on the participants’ performance (as presented by their correct answers and completion time).

The variables “Same score” and “Same time” referred to the score (correct answers to each test) and the time (in seconds) participants needed to complete each trial, respectively, and those surfaces for which the variables “friction” (test 1) or “hardness” (test 2) had the same value (see, for instance, Figure 2 upper-left and upper-right for friction, and upper-left and bottom-right for hardness). The variables “Dif.1 score” and “Dif.1 time” referred to the score and the completion time, respectively, for those surfaces of each test that differed to each other slightly in terms of the values of the variables “friction” or “hardness” (see, for instance, Figure 2 upper-left and bottom-right for friction and upper-left and upper-right for hardness). The variables “Dif.2 score” and “Dif.2 time” referred to the score and the completion time, respectively, for those surfaces of each test that differed to each other a lot in terms of the values of the variables “friction” or “hardness” (see, for instance, Figure 2 bottom-left and bottom-right for friction and upper-right and bottom-left for hardness). Figure 4 presents the results for the participants as a total (participants with and without VI).

A two-way mixed-design ANOVA with one between-subjects measure and one repeated measure was implemented, both for the mean score of the correct answers and the mean completion time; the vision status (visually impaired vs. sighted) was the between-subjects variable, and the comparison pair type (Same, Dif.1, and Dif.2) was the within-subjects variable.

Regarding the mean scores for the “friction” variable, the analysis did not reveal any significant main effect for the comparison pair type and vision status. Moreover, the analysis did not reveal any significant interaction between the comparison pair type and vision status either.

Regarding the mean completion time for the “friction” variable, the analysis revealed a significant main effect for comparison pair type (F(1.500, 127.480) = 16.578, *p* < 0.01) and a significant main effect for vision status (F(1, 85) = 7.208, *p* < 0.01). The analysis did not reveal any significant interaction between the comparison pair type and vision status. Mauchly’s test of sphericity indicated that the assumption of sphericity had been violated and a Greenhouse–Geisser correction was therefore used. In addition, implementation of the Bonferroni post-hoc test yielded significant differences between the variables “Same time” and “Dif.1 time” (*p* < 0.01), as well as between the variables “Same time” and “Dif.2 time” (*p* < 0.01); however, significant differences between the variables “Dif.1 time” and “Dif.2 time” were not observed.

More specifically, in trials where the surfaces differed to each other slightly or a lot in terms of the values for “friction”, participants presented lower completion time compared to trials where the surfaces were the same in terms of the value for “friction”. Moreover, participants with VI presented higher completion time compared to sighted participants but did not differ from each other in terms of the mean scores for correct answers.

Regarding the mean scores for the “hardness” variable, the analysis revealed a significant main effect for comparison pair type (F(1.667, 141.663) = 6.392, *p* < 0.01) and a significant main effect for vision status (F(1, 85) = 5.781, *p* < 0.05). The analysis did not reveal any significant interaction between the comparison pair type and vision status. Mauchly’s test of sphericity indicated that the assumption of sphericity had been violated and a Greenhouse–Geisser correction was therefore used. In addition, implementation of the Bonferroni post-hoc test yielded significant differences between the variables “Dif.2 score” and “Same score” (*p* < 0.01), as well as between the variables “Dif.1 score” and “Dif.2 score” (*p* < 0.01); however, no statistically significant differences between the variables “Dif.1 score” and “Same score” were observed.

Concerning the mean completion time for the variable “hardness”, the analysis revealed a significant main effect for the comparison pair type (F(1.853, 157.512) = 3.973, *p* < 0.05) but no significant main effect for the variable vision status. The analysis did not reveal any significant interaction between the comparison pair type and vision status either. Mauchly’s test of sphericity indicated that the assumption of sphericity had been violated and a Greenhouse–Geisser correction was therefore used. In addition, the Bonferroni post-hoc test revealed significant differences between the variables “Same time” and “Dif.2 time” (*p* < 0.05). Neither between the variables “Same time” and “Dif.1 time” nor between the variables “Dif.1 time” and “Dif.2 time” were any significant differences revealed.

More specifically, for surfaces that differed a lot from each other in “hardness”, the participants presented higher scores compared to their scores for surfaces which did not differ from each other (the same surfaces) or for surfaces which differed from each other slightly in terms of “hardness”. In addition, with reference to surfaces that differed a lot from each other in terms of “hardness”, the participants presented lower completion time compared to the pairs that did not differ from each other (the same surfaces) in terms of “hardness”. Finally, participants with VI presented lower scores compared to sighted participants, but they did not present higher completion time.

In addition, a t-test analysis was conducted to examine the possible impact of individual characteristics on participants’ performance. The analysis yielded no statistically significant difference regarding participant performance between (a) male and female participants either with VI or without VI, (b) between participants with blindness and participants with residual vision, or (c) between Braille readers and participants who did not use Braille or used it rarely. Moreover, correlation analysis did not reveal any statistical significance between performance and age at onset of VI or age of vision loss.

## 4. Discussion

The present research examined the ability to perceive differences in two basic properties of objects’ surfaces—friction and hardness—by individuals with and without VI. The same and different surfaces were compared in terms of the exponents of friction and hardness using a force feedback device.

### 4.1. Better Performance in Different Surfaces

According to the results, all the participants spent more time examining surfaces that were the same as far as the exponent of friction was concerned. On the contrary, surfaces that differed in terms of the friction exponent were more or less easy for the participants. This means that when friction is being examined as a 3D surface property, differentiation is necessary for time saving during exploratory tasks. Similarly, Ziherl and Munih [34] found that rendering materials in a virtual environment using the friction model implies identification and difficulties, and therefore suggested exaggerating friction properties in order to enable the user.

Furthermore, when hardness is being examined, a high exponent of differentiation is needed for the most efficient and time-saving perception of the 3D stimuli. In the present study, with reference both to their score and the time needed to complete the trial, participants performed better when the surfaces presented the greatest differentiation in hardness. This is in line with the results of previous studies. It has been suggested that when the stiffness of a surface is maximized the vibratory frequency is raised [29,30]. Consequently, the feedback (i.e., the intensity of stimulation) on the user’s receptors (skin, fingers, etc.) is increased and, as a result, the user perceives hardness more easily [31].

### 4.2. The Need to Scale Haptic Properties

As in the present research, when Hollins and his colleagues [18] examined the perception of stiffness (hardness) using various exponents they arrived at the same exponent (1.01) and found that there was a clear effect caused by stiffness differentiation size in terms of the participants’ understanding of the surfaces. The result of the study mentioned above that deviates from the present research is the exponent’s size for friction. Hollins et al. [18] suggested a 1.49 exponent for stickiness (friction), while in the present study both 0.35 and 0.70 were effective when they were compared to different exponent-size surfaces. A hypothesis that could be formulated with certainty is that Hollins et al. [18] followed a research design where the participants were allowed specific exploratory movements, while in our case participants were not restricted. In addition, they themselves found that the change in exploratory behavior led to different results.

When discussing friction and hardness in combination, the main conclusion is that both need a degree of exponent differentiation between similar surfaces with respect to the range of values in each case. Hardness, especially, needs the greatest possible differentiation as it appears to create more difficulties in haptic perception. Of course, a surface’s formation or geometry could have an impact on the comprehensibility of the virtual object, and thus create the need for a different friction and/or hardness exponent. According to Martínez et al.’s study [8], when the 3D stimulus was not a geometric shape, subjects scanned the surface at a constant speed to examine the frequency of the marks and perceive the texture. This means that, based on vibration, the subject can acquire the necessary information. If this is the case then someone could conclude that it is vibratory feedback which differentiates friction from hardness. Hardness seems to lack the appropriate vibratory feedback and that is probably why it needs the maximum possible exponent. Heller [10] observed in his study that, in smoother surfaces, individuals changed their exploratory study by pushing their index against the surface, probably in an attempt to maximize the vibratory or frictional sense. Hollins and his colleagues [18] actually proved Heller’s [10] observation with virtual surfaces. They found that, when participants were permitted to press down firmly on the surfaces, the magnitude of property estimation was higher than estimations performed after gentle pressure. Thus, in cases where a subject encounters both (a) haptic exploration through a force feedback device (which by definition entails identification restrictions since the contact with the stimulus is mediated by a stylus) and (b) a stimulus (such as a hard surface that does not provide enough vibratory or frictional cues), identification becomes more difficult and time consuming compared to surfaces with ample vibratory feedback.

### 4.3. Haptic Exploration by Individuals with VI

Moreover, specific attention should be directed to the case of users of force feedback devices with VI. When 3D objects are prepared for haptic exploration by users with VI, more time should be available to them to identify friction compared to the time users without VI might need. In the research of Jansson et al. [22], four out of the ten participants had difficulties completing the experiment, though these participants improved their performance after training. As far as hardness is concerned, users with VI faced more difficulties in identifying surfaces with hardness differentiation and, thus, a great hardness exponent should be applied on surfaces. In line with this result, Hollins et al. [18] found that when vision was obstructed, exploratory behavior was altered. Additionally, motor behavior was variable, though downward force exerted was increased because of the difficulty of distinguishing the stimulus.

## 5. Conclusions

The present study aimed to examine two significant properties of 3D haptic surfaces—friction and hardness—by comparing the same and different surfaces with regard to their exponent. Specifying the degree of differentiation for individuals with and without VI will enable the development of 3D objects and models that are easily identifiable in terms of their detail and thus can be effectively used. For instance, statues in museums could be rendered virtually, and visitors with and without blindness could access them by touching the facial features. Cultural and popular tourist destinations could provide 3D models for orientation which would elevate their level of accessibility. In addition, accessible educational material could be developed for students with severe VI so that they receive support for the textual information they receive with the figurative content. To conclude, accessibility advisors and experts could definitely exploit the results of the present study.

It should be noted that to achieve the standardization of 3D haptic objects for individuals with and without VI, a more holistic approach should be followed. This means that exponent escalation could be examined in more detail, while other variables/surface properties should be studied alone or in combination with each other. Future research should focus on this.

## Figures and Tables

**Figure 1 sensors-22-09745-f001:**
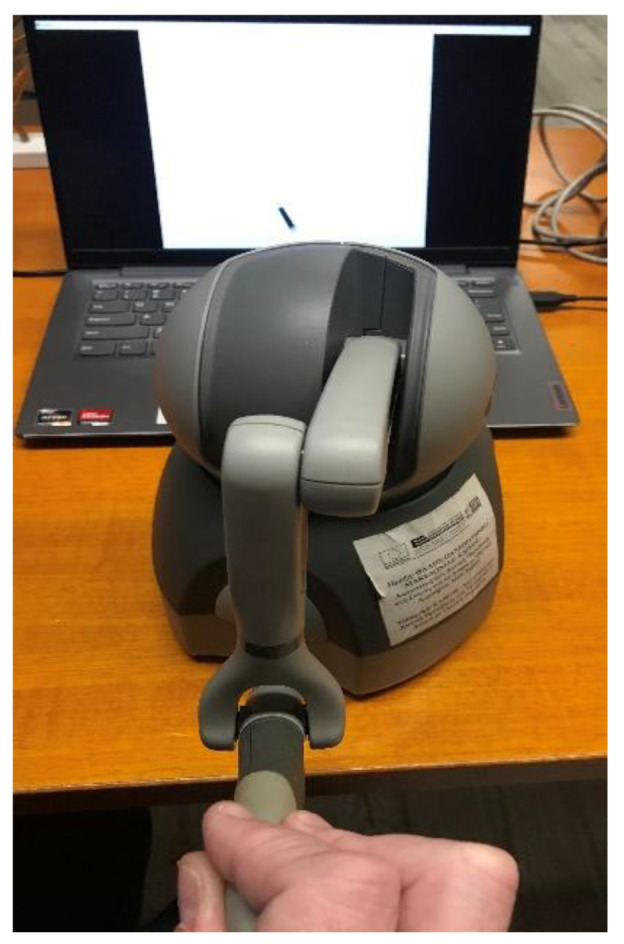
A participant sitting in front of a portable computer using the Touch.

**Figure 2 sensors-22-09745-f002:**
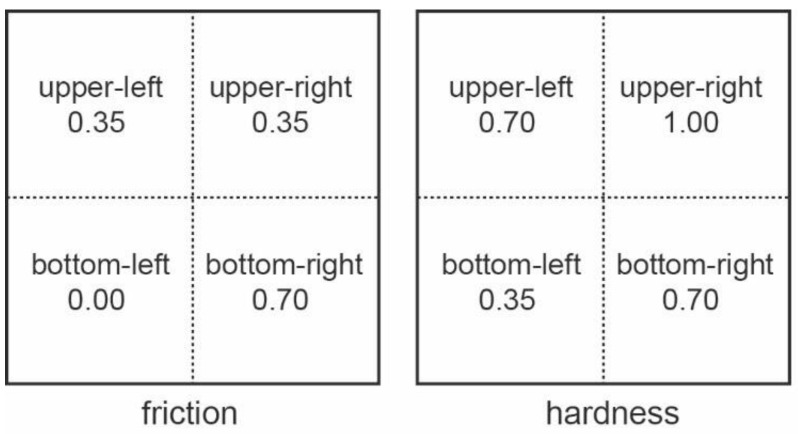
The two variables—friction and hardness—are represented in the two tests. Each test is divided into four virtual surfaces. There is a specific value for each surface, but two out of the four surfaces had the same values.

**Figure 3 sensors-22-09745-f003:**
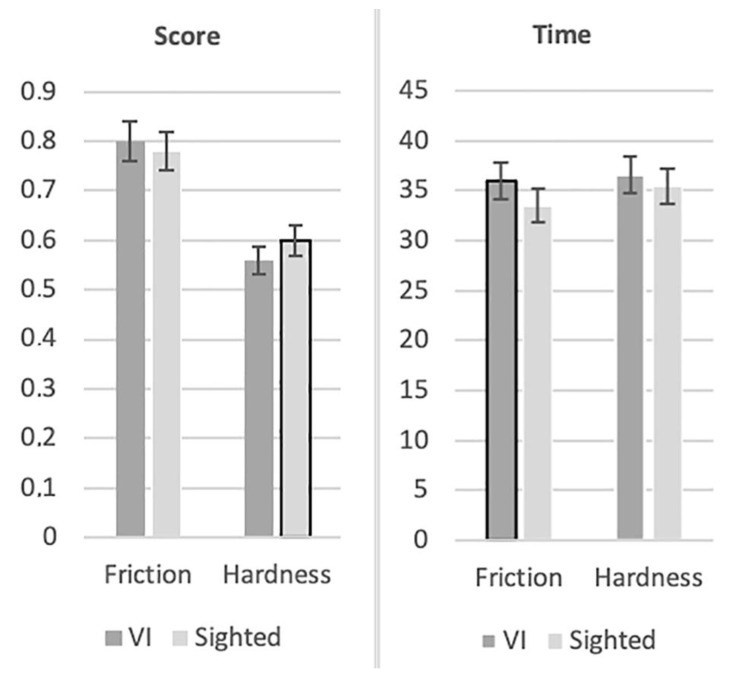
Participants’ scores (participants with VI and sighted participants) and their completion time (seconds) in each test. The significant differences are highlighted with a black border.

**Figure 4 sensors-22-09745-f004:**
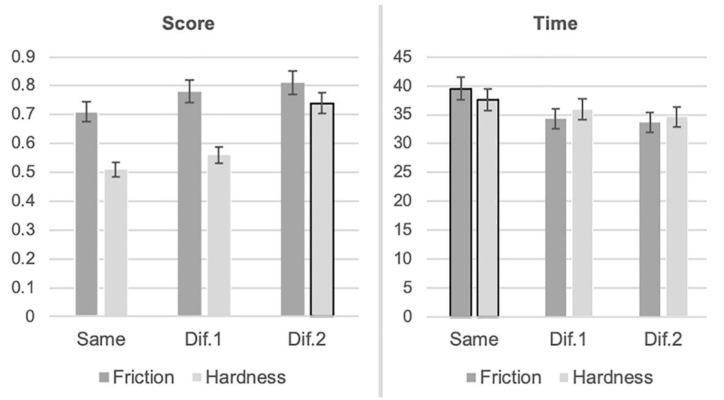
Participants’ score and completion time for those surfaces that had the same value of friction or hardness or surfaces that differed from each other slightly (Dif.1) or a lot (Dif.2) in terms of the value of friction or hardness. The significant differences are highlighted with a black border.

## Data Availability

The data presented in this study are available on request from the corresponding author. The data are not publicly available due to privacy restrictions.

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
