# Peer review of "Individuals with and without Visual Impairments Use a Force Feedback Device to Identify the Friction and Hardness of Haptic Surfaces"

_sensors, 2022, doi:10.3390/s22249745_

Round 1

Reviewer 1 Report

In general, the paper is well written.

I have only one concern regarding the statistical analysis. The authors used an ANOVA to analyze the time. However, time measurements are usually not normally distributed. I suggest authors reanalyze the data using a general linear model that takes into account the shape of the reaction time distribution. Usually, the shifted lognormal distribution is the distribution that better approximates time data (https://lindeloev.shinyapps.io/shiny-rt/).

Additionally, since they have an accuracy score, they could analyze the search time with the response with a drift-diffusion model. In this case, they should have a measurement of evidence accumulation before reaching a decision in different groups and comparing it between groups. It may be that VI people need less evidence than sighted individuals to perform the task accurately.

I suggest displaying the results on a figure rather than a table. It is much more immediate to grasp the results with a figure rather than a table. 

Reviewer 2 Report

In the work presented in the manuscript, a series of experiments focusing on the discrimination of different surfaces with different properties (i.e., hardness and roughness) rendered virtually using force feedback devices. The experiment compared also the performance in participants with and without visual impairments. They found that discriminating surfaces with different properties is easier than discriminating surfaces with similar hardness or friction. They said also that difference in hardness were difficult to discriminate with respect to difference in friction. Difference in discrimination time and score was identified for sighted and participants with visual impairments. The theme of the manuscript is in line with the journal. The work involved also an high number of participants with visual impairments and no. However, many information are not explained, and other parts of the manuscript are not clear. Figures relative to results and setup should be added. See points below.

·       In abstract a little summary of the results should be presented.

·       In the introduction, the authors talked about three dimensions and specific features of the haptic perception: roughness, hardness and stickiness. Roughness and stickiness should define together the friction. However, the sentence “As regards the third dimension of surface haptic perception, stickiness and slipperiness are related to the dimension of friction [19]. Few studies [e.g., 18,33,34] have investigated the perception of friction using haptic feedback devices, in particular, the Phantom device used by participants without VI in these studies” seems to suggest that friction seems to related to mainly on stickiness. In general the difference between all these properties should be better explained.

·       The authors wrote about difference in exponents in introduction and discussion. It is not clear what this exponent is. Are these numbers related to physical parameters (for example, ratio between tangential and normal force for friction)? For the use of such kind of parameters, did the authors refer to one or more reference? The authors compare the findings of hardness and friction, but are these exponents really comparable for the different properties?

·       The sentence “The textures’ differentiation could be based on the use of different haptic variables or the use of the same haptic variable but of different values so that the facial features could be sensed differently” is not so clear probably the authors should provide examples to better explain the meaning.

·       The authors wrote that each test is divided into four surfaces. These surfaces are presented in fixed positions of the test and the test is presented 6 times. It is possible that the participants can identify the surface using the information relative to the position of the surface after several times that participant performed the task. The authors should better discuss such point

·       In Results the authors should better explain what it is the score that they use (the ratio of correct responses with respect to the number of trials).

·       The presentation of the data in the tables with figures could make the result data more intuitive to understand

·       When the authors wrote about the Same, Dif.1 and Dif.2 score should provide the group of surface comparisons that belong to each group.

·       I would not use the “variable type” as name of one factor, the use of “comparison pair type” as factor name could be more correct.

·       In rows 252- 255 the authors cited significant difference between Dif.2 and Same scores and no significant difference between Dif.1 and Dif.2 score, but nothing is written about difference between Same and Dif.1 score

·       In Instruments and procedures, I suggest to present early a sentence that describes generally the system employed in the experiment, such as “We used a system constituted by a haptic feedback device with three degree of freedom and six degrees of freedom of positional sensing, the system integrates a pen-like handle as end effector. Thanks to a simulation software, it is possible to create an interaction for the user with a virtual surfaces with different properties”. After such explanation the authors could  which kind of device was selected and why.

·       A photo or image that presents the setup could be useful for better understand the task.

·       The sentence “when stiffness is maximized, the perception of hardness follows because of the vibratory frequency [29,30], and when the cutaneous feedback is increased the perception of hardness is supported accordingly [31].” is generally not clear and should be re-written. In particular, what did the authors mean with cutaneous feedback?

Round 2

Reviewer 2 Report

The manuscript resulted improved after the changes driven by the reviewers’ comments. I have few additional comments:

The sentence in abstract: “The results showed that it is necessary to scale the haptic properties since different surfaces are better perceived” is not clear. Please re-write the sentence.

In introduction, in the sentence “it has been suggested that only specific features of the stimulus can be perceived, such as vibration and roughness” it should be specified that such features can be extracted only by moving the handle on the surface.

The figures don’t include the errorbars, also the comparisons with significant difference could be highlighted.

The photo or image of the setup could include the position of the subject in front to the haptic device and how he/she manipulates such device.

In the article the authors reported that “When discussing friction and hardness in combination, the main conclusion is that both need a degree of exponent differentiation between similar surfaces, but hardness needs the greatest possible differentiation as it appears to create more difficulties in haptic perception” but, in general, the different exponents of hardness and friction are not comparable (difference in exponents could have different scales for the hardness and friction), what happens here could be related to the specificity of the selected surfaces and little generalizable, please add a sentence about it.

In the sentence “.. their exponent, i.e. the value that defines the degree of friction or hardness” the authors did not define what physical are these exponent. Please add more details.

In the sentence “..Consequently, the feedback on the user’s receptors (skin, fingers etc.) is increased and as a result the user perceives hardness more easily” what the authors intend with the feedback? The quantity of information that can provided to the receptors? The intensity of the stimulation?

The  sentence “Like the present research, when Hollins and his colleagues [18] examined the perceptual scaling of stiffness (hardness) property in virtual surfaces, they arrived at the same exponent (1.01), and found that there was a clear effect of the stiffness differentiation size in judging the surfaces’ difference.” is hard to understand. Please re-write it.

Author Response

Dear Reviewer,

We would like to thank you for the time you have spent to our paper and for the opportunity to improve the quality of it. Please, find attached our responses to your comments.

Kind regards,

The authors
